# The Effect of the Stringent Response and Oxidative Stress Response on Fitness Costs of De Novo Acquisition of Antibiotic Resistance

**DOI:** 10.3390/ijms25052582

**Published:** 2024-02-23

**Authors:** Wenxi Qi, Martijs J. Jonker, Drosos Katsavelis, Wim de Leeuw, Meike Wortel, Benno H. ter Kuile

**Affiliations:** 1Laboratory for Molecular Biology and Microbial Food Safety, Swammerdam Institute for Life Sciences, University of Amsterdam, Science Park 904, 1098 XH Amsterdam, The Netherlands; w.qi@uva.nl (W.Q.); d.katsavelis@rug.nl (D.K.); m.t.wortel@uva.nl (M.W.); 2RNA Biology & Applied Bioinformatics, Swammerdam Institute for Life Sciences, University of Amsterdam, Science Park 904, 1098 XH Amsterdam, The Netherlands; m.j.jonker@uva.nl (M.J.J.); w.c.deleeuw@uva.nl (W.d.L.)

**Keywords:** de novo antibiotic resistance, fitness cost, (p)ppGpp, reactive oxygen species, compensatory evolution

## Abstract

Resistance evolution during exposure to non-lethal levels of antibiotics is influenced by various stress responses of bacteria which are known to affect growth rate. Here, we aim to disentangle how the interplay between resistance development and associated fitness costs is affected by stress responses. We performed de novo resistance evolution of wild-type strains and single-gene knockout strains in stress response pathways using four different antibiotics. Throughout resistance development, the increase in minimum inhibitory concentration (MIC) is accompanied by a gradual decrease in growth rate, most pronounced in amoxicillin or kanamycin. By measuring biomass yield on glucose and whole-genome sequences at intermediate and final time points, we identified two patterns of how the stress responses affect the correlation between MIC and growth rate. First, single-gene knockout *E. coli* strains associated with reactive oxygen species (ROS) acquire resistance faster, and mutations related to antibiotic permeability and pumping out occur earlier. This increases the metabolic burden of resistant bacteria. Second, the Δ*relA* knockout strain, which has reduced (p)ppGpp synthesis, is restricted in its stringent response, leading to diminished growth rates. The ROS-related mutagenesis and the stringent response increase metabolic burdens during resistance development, causing lower growth rates and higher fitness costs.

## 1. Introduction

Antibiotic resistance has become one of the major public health threats of the 21st century [1]. Overuse or misuse of antibiotics in hospitals, communities, and farms has led to the emergence, spread, and persistence of resistant strains, reducing the effectiveness of the prevention and treatment of bacterial infections, in particular through gram-negative pathogens [2,3]. Understanding the mechanisms of the de novo acquisition of antibiotic resistance is critical for the discovery of novel drug targets and the development of new antibiotics that will remain effective during their application. In addition, studying the physiological and metabolic costs of the evolution of resistance may identify manners to reduce resistance by designing new therapies targeting physiological weaknesses associated with specific resistance mechanisms [4,5].

The development of antimicrobial resistance, using both de novo acquisition and the horizontal transfer of resistance genes, is frequently accompanied by a decline in bacterial fitness [6,7,8]. Conversion of cellular homeostasis systems, such as ion- and pH-pumps, into antibiotic efflux pumps causes a reduced capability to control the intracellular environment [9]. Mutations within genes associated with resistance, such as the modification of drug targets, degradation of antibiotics, reduced antibiotic uptake, and increased efflux, can result in diminished survival and replicative capabilities among resistant populations [4]. Consequently, it has been suggested that limiting antibiotic usage, and, thus, reducing the selection of pressure-favoring resistant populations, may allow more adaptable and susceptible populations to outcompete and, ultimately, eradicate resistance [10]. However, several other responses of bacteria becoming resistant, such as the co-selection of resistance genes alongside other functional traits, can provide an adaptive advantage [11]. The acquisition of resistance plasmids did not reduce growth rates in *E. coli* [12]. Low-cost resistance mutations and compensatory evolution can compensate for the initial metabolic costs [10]. Therefore, a comprehensive understanding of the intricate relationship between mutation occurrence and fitness costs is essential for addressing the persistent challenge of antibiotic resistance.

Under antibiotic stress, bacteria initiate a variety of stress responses such as stringent and oxidative stress responses [13]. Bacterial stringent and oxidative stress responses are involved in the development of antibiotic resistance [14,15]. Bactericidal antibiotics interact with their targets in a way that increases the oxidation of NADH and thus produces respiratory chain byproducts such as reactive oxygen species (ROS) [16]. Although ROS at high concentrations kills bacteria, DNA damage caused by sub-lethal levels of ROS can activate the repair system induced by the SOS response, thereby increasing the mutation rate [14,17]. Ultimately, exposure to antibiotics can cause mutations in antimicrobial resistance-related genes. The single-gene knockout strain of (p)ppGpp synthase RelA has a reduced stringent response, resulting in lower growth rates and reduced ROS formation, reducing the development of drug resistance [15]. As these two stress responses also affect the growth rate, it is unclear to what extent the fitness costs during the development of resistance are related to the stress responses.

Important genes in the (p)ppGpp-related stringent response include *hipA*, *hipB*, *relA*, and *rpoS*. HipA and HipB are part of the type II toxin–antitoxin system and can affect glutamate-tRNA ligase [18]. HipA phosphorylates glutamate-tRNA ligase, causing uncharged tRNA (Glu) accumulation, and thereby hindering the translation process [19]. The reduced translation rates increase (p)ppGpp levels, which, catalyzed by RelA/SpoT, activates the stringent response. As a result, RNA polymerase switches from transcribing growth and reproduction-related genes to stress-response-related genes, and (p)ppGpp accelerates amino acid biosynthesis [20]. HipB acts as an antitoxin component and can neutralize the toxicity of HipA [21]. Therefore, knocking out *hipA* or *hipB* may affect the stringent response induced by (p)ppGpp. RelA is a GTP pyrophosphokinase, that catalyzes the formation of (p)ppGpp, and a knockout of *relA* most likely directly affects the synthesis of (p)ppGpp [22]. (p)ppGpp binds to RNA polymerase and responds to stress through the sigma factor σ^S^ [23]. Therefore, directly knocking out the gene *rpoS*, encoding RNA polymerase sigma factor σ^S^, has a potential impact on stress response. In the system dealing with ROS, *sodA*, *sodB*, *soxR*, and *soxS* play important roles. Superoxide dismutase SodA and SodB can destroy superoxide anion radicals [24]. Therefore, after knocking out these genes, the function of clearing excessively generated ROS in cells will be weakened to a certain extent. The gene *soxR* encodes a redox-sensitive transcriptional activator responsible for triggering the transcription of the superoxide response regulator SoxS, which plays a role in the removal of superoxide [25]. Consequently, the deletion of *soxR* or *soxS* can reduce the clearance of ROS.

In this study, we evaluate the correlation between changes in minimum inhibitory concentration (MIC) and growth rate as an indicator for fitness throughout the resistance acquisition process. For this assessment, four single-gene knockout *E. coli* strains related to the stringent response were used; moreover, four were related to the oxidative stress responses. The fitness costs associated with resistance were investigated for each strain by quantifying the biomass yield on glucose. Evidence for the relationships between mutations and fitness was ultimately discovered through whole-genome sequencing. The overall picture that emerges is that initial mutations causing resistance come at a metabolic cost that can be subsequently compensated using compensatory mutations. The complex cellular response to stress due to exposure to antibiotics, mediated by the stringent response, oxidative stress response, and other cellular processes, allows the cell to overcome non-lethal concentrations of antimicrobials at minimal metabolic costs.

## 2. Results

### 2.1. Growth Rates Gradually Decreased during the Evolution of Resistance

To investigate the effect of the stringent response and oxidative stress response on the fitness cost of resistant strains during the de novo acquisition of antibiotic resistance, growth rates were measured at each MIC value reached. Four (p)ppGpp-related (Δ*hipA*, Δ*hipB*, Δ*relA*, and Δ*rpoS*) and four ROS-related (Δ*sodA*, Δ*sodB*, Δ*soxR*, and Δ*soxS*) single-gene knockout *E. coli* strains and the wild-type MG1655 strain were exposed to stepwise increasing levels of amoxicillin, enrofloxacin, kanamycin, and tetracycline. When these strains evolved higher resistance, the growth rate in an antibiotic-free medium and the MIC were recorded, and it showed that, generally, the growth rate decreased with increasing MIC (Figure 1). Before the resistance evolution, the growth rates of all ancestor strains were tested, and no significant differences were found (Figure 1Q).

The growth rates during the amoxicillin resistance evolution of Δ*hipA* and Δ*hipB* were consistently similar to that of the wild type (WT) (Figure 1A). At the early stage of amoxicillin resistance evolution, when the MIC was 8 µg/mL, the growth rates of Δ*relA* and Δ*rpoS* were clearly lower than WT; however, when the MIC reached 2048 µg/mL, the growth rate of Δ*rpoS* was equal to that of WT (Figure 1E). For ROS-related knockout strains, the growth rate began to decrease at the middle stage of the evolution when the MIC was 128 µg/mL but was similar to WT at the highest concentration (Figure 1I,M). When the MIC reached 128 µg/mL, the growth rates of Δ*relA*, Δ*sodB*, Δ*soxR*, and Δ*soxS* were significantly reduced compared with WT (Figure 1R). The growth rate of all amoxicillin-resistant strains decreased gradually with an increase in MIC. Among them, Δ*soxR* and Δ*soxS* decreased the fastest, and their slopes were 0.025 and 0.030, respectively, compared to 0.018 for the WT (Figure 1M).

The evolution process of enrofloxacin resistance was relatively complicated. Some strains such as Δ*hipA*, Δ*relA*, and Δ*rpoS* cannot evolve to higher enrofloxacin concentrations, which makes them difficult to compare (Figure 1B,F). The other knockout strains had no significant difference compared with WT (Figure 1B,J,N). However, enrofloxacin-resistant strains showed a lower slope of decline in growth rate than amoxicillin throughout the evolution.

During kanamycin resistance evolution, Δ*hipA* and Δ*hipB* had a higher growth rate than WT until reaching the middle phase, i.e., until the MIC reached 256 µg/mL (Figure 1C). Similarly, the growth rates of Δ*sodA*, Δ*sodB*, Δ*soxR*, and Δ*soxS* were slightly higher than those of WT at MICs lower than 128 µg/mL (Figure 1K,O). However, their growth rates decreased significantly thereafter and reached the same level as WT at the highest concentration. Similar to the case of amoxicillin, the growth rate of Δ*relA* during the kanamycin resistance evolution was lower than that of the WT starting from MIC 16 µg/mL (Figure 1G). Δ*rpoS* showed a clear and gradual decreasing trend with an R-squared value of 0.957, indicating a high correlation between a decrease in growth rate and an increase in MIC (Figure 1G). Compared with WT, the growth rates of Δ*relA*, Δ*rpoS*, Δ*sodA*, Δ*sodB*, Δ*soxR*, and Δ*soxS* were significantly lower than that of WT when the MIC was 256 µg/mL (Figure 1S). The growth rate reduction slopes of the kanamycin-resistant evolution were steeper than those of amoxicillin, respectively, indicating that it has a higher fitness burden.

The growth rate during the evolution of tetracycline resistance was not significantly different from that of WT and either (p)ppGpp-associated or ROS-associated mutants (Figure 1D,H,L,P). The highest MIC evolved by all the *E. coli* strains to the bacteriostatic antibiotic tetracycline was 128 µg/mL, which was lower than 2048 µg/mL or 4096 µg/mL for the bactericidal antibiotics. For all strains, it is difficult to evolve very high resistance to tetracycline, limiting the comparison of growth rates between the strains.

### 2.2. Decrease in Biomass Yield on Glucose with Increased Resistance

A clear correlation between the growth rate and MIC was observed during the amoxicillin or kanamycin resistance evolution, and both treatments’ growth rates showed a significant drop-down at the middle stage of the resistance evolution in specific mutants compared to WT (Figure 1). Such correlation was far less clear, if at all present, in the exposed enrofloxacin and tetracycline strains. Not all enrofloxacin-exposed mutants were able to evolve high levels of resistance. Tetracycline is a bacteriostatic antibiotic and was used as a biological control to compare with the bactericidals. Therefore, further analysis concentrated on the amoxicillin and kanamycin exposed and evolved strains.

To further understand the impact of the stringent response and oxidative stress response on fitness costs during antimicrobial resistance evolution, we measured biomass yield on glucose during amoxicillin or kanamycin resistance evolution in three stages, namely, the initial, middle, and end stages (Figure 2). These experiments were terminated at OD = 1.0, while the incubations in the evolution experiments were almost always continued until the stationary phase.

Firstly, there was no significant difference between the ancestor WT MG1655 and the single-gene knockout mutants before resistance evolution (Figure 2). At the middle stage (MIC = 128 µg/mL) of amoxicillin resistance evolution, the biomass yield of Δ*relA*, Δ*sodB*, Δ*soxR*, and Δ*soxS* were significantly lower than that of the WT MG1655 (Figure 2A). Additionally, compared with the final resistant point of WT MG1655, the biomass yield of Δ*relA* was significantly decreased (Figure 2A). 

Except for Δ*hipA* and Δ*hipB* in the middle stage of the kanamycin-resistant evolution, all the other mutants showed a lower biomass yield than that of the WT (Figure 2B). Similar to the amoxicillin resistance evolution, only the biomass yield of Δ*relA* was significantly decreased compared to WT at the kanamycin evolution endpoint. The biomass yield of all final resistant strains was lower than their ancestor strains and the middle stage of the evolution-resistant strains, respectively (Figure 2A,B). In summary, during the evolution of amoxicillin or kanamycin resistance, with the increase in resistance levels, the growth rate of specific (p)ppGpp- or ROS-related resistant strains gradually decreased, accompanied by decreased biomass yield on glucose.

### 2.3. Mutations in the Amoxicillin-Resistant Strains during the Resistance Evolution

To detect the mutations and determine the correlation between fitness cost and mutations upon the development of antibiotic resistance, the whole genome of all amoxicillin or kanamycin-resistant strains in the ancestor, middle, and end resistance points were sequenced. The shared mutations between the resistant strains at the middle or end evolution points and their respective ancestor strains were excluded. The population frequency of each mutation is represented by bars of different lengths (Figure 3 and Figure 4).

All amoxicillin-resistant strains contained mutations in the *ampC* promoter region (Figure 3; Table 1). Most of these mutations appear in the middle stage and are preserved until the final stage. Some appear in the middle stage and disappear in the final stage, and some only appear in the end stage. The difference in these mutations may be a result of the selection made for the different RNA polymerase recruitment abilities to retain the most efficient increase in transcription levels, ultimately increasing the expression of beta-lactamase AmpC [26]. Mutations occurring in the *ampC* promoter region are common among all strains; therefore, their effects on fitness costs cannot be elucidated from these data.

Several other mutations emerged in different strains during the evolution of amoxicillin resistance (Figure 3). Resistance mechanisms involved in these mutations include reduced antibiotic permeability, increased antibiotic pumping, and altered antibiotic targets. Notably, in the middle of evolution, a *cpxA* mutation emerged in Δ*hipA*, Δ*sodA*, and Δ*soxR*. In addition, other mutations in *cpxA*, albeit in different positions, also occurred in a high population frequency at the Δ*sodB* and Δ*soxS* middle point. Sensor histidine kinase CpxA initiates a kinase cascade leading to the activation of CpxR, which, subsequently, functions to enhance the expression of efflux complexes [27]. At the middle point of Δ*sodB*, other efflux pump-associated genes also mutated. Some other notable mutations at the middle point were *envZ* (*ompB*) in Δ*sodB* and Δ*soxS*, and *ompC* in Δ*soxR*. These mutations may be related to reduced antibiotic permeability [28,29]. To sum up, mutated genes related to antibiotic permeability and efflux pumps appeared simultaneously at high frequency in Δ*sodB*, Δ*soxR*, and Δ*soxS*. Compared to the wild type, these three strains had lower biomass yields at the middle stage of evolution (Figure 2A). This suggests that bacteria have evolved mutations that reduce antibiotic permeability and increase efflux, thereby increasing their energy expenditure burden.

At the endpoint of the evolution experiments, almost all the amoxicillin-resistant strains evolved mutations associated with reduced antibiotic permeability except Δ*relA* (Figure 3). Surprisingly, only Δ*relA* had a significantly lower biomass yield on glucose at the endpoint than WT, while there were no obvious differences between the other strains (Figure 2A). Consistent with the biomass yield results observed at the midpoint, we hypothesize that, in Δ*relA*, the effect of glucose consumption due to the mutation is not as large as the effect of *relA* knockout itself. This phenomenon may be caused by the production of (p)ppGpp, which is inhibited to a certain extent after the deletion of *relA*. Along with the mutation’s side effects, stringent responses are hindered. The consequence of this is that internal signals are configured as if the cells are in a state of constant starvation [30,31].

### 2.4. Mutations in the Kanamycin-Resistant Strains during the Resistance Evolution

The common mutated gene of all kanamycin-resistant strains observed at the middle stage was *fusA*, which codes for the elongation factor G (EF-G) (Figure 4) (Table 2). FusA catalyzes the ribosomal translocation step during translation elongation, and the mutations in *fusA* may avoid kanamycin binding with EF-G and prevent translation [32]. Similar to *ampC* mutations in amoxicillin resistance, kanamycin-induced *fusA* mutations are drug target mutations that occur in every resistant strain; therefore, their effects on fitness costs cannot be compared. 

At the middle point of evolution, WT MG1655, Δ*rpoS*, and Δ*sodA* developed *sbmA* mutations (Figure 4). *sbmA* encodes a peptide antibiotic transporter, and mutations in this gene may reduce the permeability of antibiotics [33]. There were other mutated genes that occurred in Δ*rpoS*, such as *ubiA*, *oppD*, and *dxs*, whose functions in antibiotic resistance were unclear; however, these mutations may be responsible for the reduced growth rate and increased glucose consumption in the middle stage (Figure 1S and Figure 2B). In addition to *sbmA*, another gene associated with reduced antibiotic uptake, *trkH* [34], was mutated in Δ*sodA*. In Δ*sodB*, *trkH* and other mutations emerged, potentially indicating a higher fitness burden than that of WT. In Δ*soxR* and Δ*soxS*, an antibiotic efflux-associated gene, *kdpD*, was mutated earlier than in other strains [35], which may also be the reason why their fitness cost is higher than that of WT. Only Δ*hipA*, Δ*hipB*, and Δ*relA*, did not have antibiotic permeability and efflux-related gene mutations at the midpoint. However, a mutation in the *spoT* gene appeared in Δ*relA*. *spoT* encodes another (p)ppGpp synthase [30]. The mutation of *spoT* potentially suggests that the bacterial stringent response to antibiotic stress is affected after knocking out *relA*. This, in turn, indicates that, unlike other strains, which increase fitness costs due to genetic mutations, the *relA* knockout strain itself has a higher fitness cost in response to antibiotic stress, and it is not specific to a certain antibiotic. The absence of mutations related to antibiotic permeability and efflux at the endpoint of Δ*relA* kanamycin resistance further illustrates this point.

## 3. Discussion

In this study, we found that, as resistance increases during amoxicillin or kanamycin evolution, the growth rate of resistant strains gradually decreases. This is to be expected because bacteria must spend the additional energy on either resistance mutations or on additional metabolic activity [9]. We tried to address the question of whether there is a common pattern for this increase in fitness cost, and whether other stress responses have synergistic or antagonistic effects. The answer we propose is that two types of growth inhibition patterns of resistant strains related to these two stress responses exist and that both the stringent response and oxidative stress response have an impact on the fitness cost during the evolution of resistance. 

The reduced growth rates of ROS-related single-gene knockout strains Δ*sodA*, Δ*sodB*, Δ*soxR*, and Δ*soxS* (Figure 5) are due to mutations related to antibiotic permeability and efflux pumps that occur halfway through the evolution process. This conclusion concurs with the observation that strains with increased levels of ROS production have a faster rate of resistance development [14]. During the evolution of bactericidal antibiotic resistance, drug-target interactions stimulate NADH oxidation within the TCA cycle-dependent electron transport chain (ETC) [16]. The excessively produced superoxide compound in the ETC damages iron–sulfur clusters through the Fenton reaction, causing hydroxyl radical formation [36]. These superoxide and hydroxyl radicals, called ROS, can damage DNA [37]. In particular, guanine on the genome is oxidized by ROS to 8-hydroxy-2′-deoxyguanosine (8-HOdG), resulting in cytosine to thymine substitutions during DNA replication [38]. However, the non-lethal dose of ROS produced during non-lethal antibiotic exposure can still damage DNA and induce the formation of mutations [39]. This mutagenic effect is caused by the activation of the cell’s damage repair function through the SOS response [17]. The upregulation of the transcription level of error-prone DNA polymerase introduces errors during DNA damage repair [40]. These low-fidelity DNA polymerases can be considered a tool used by bacteria to increase their environmental adaptability by increasing their mutation rate [41]. Mutations beneficial to antibiotic resistance are ultimately retained through selection. The single-gene knockout of superoxide dismutase SodA or SodB, or superoxide response regulation, SoxR, or SoxS reduces the ability of cells to remove ROS but increases the speed of resistance acquisition [14]. These earlier-evolved mutations related to antibiotic permeability and efflux pumps all cost the cell more energy, ultimately leading to increased glucose consumption and reduced growth rate. This is, therefore, a general cellular reaction, similar in character but not identical to the ROS-mediated secondary killing mechanism of bactericidal antimicrobials. 

The growth rates of single-gene knockout strains related to the stringent response during the evolution process did not show any common features. This may be related to the different extents in which these strains trigger the alarmin (p)ppGpp under stringent responses. Only Δ*relA* showed significant changes in both growth rate and biomass yield during the evolution of amoxicillin and kanamycin resistance. When *relA* is deleted, synthesis of (p)ppGpp is reduced, and the amino acid starvation caused by the synthesis of resistance-related proteins cannot be offset in a timely manner [42]. The result is that deacylated tRNAs bind to ribosomes and hinder the translation elongation process [22]. Bacteria cannot promptly upregulate amino acid synthesis or proteolysis to increase the level of aminoacylated tRNA [43]. This will, ultimately, lead to reduced growth and increased energy requirements [44]. Moreover, the fitness burden caused by *relA* knockout is higher than that caused by specific resistance mutations. As a consequence, the growth rate of Δ*relA* decreases at lower MIC levels than the growth rate of the WT (Figure 5A,B).

The wild-type and all single-gene knockout strains used in this study were derived from the *E. coli* K12 strain of the same genetic background. Although some minor genetic differences exist between strains, no reports link these differences to resistance and fitness costs. Two typical types of gene mutations were observed in antibiotic-resistant strains: target-specific mutations, such as *ampC* upstream mutations in amoxicillin-resistant strains and *fusA* mutations in kanamycin-resistant strains; and non-target specific mutations, which mainly affect the intracellular antibiotic content, for example, reducing antibiotic uptake and increasing efflux. The mutations in the promoter region of *ampC* increase beta-lactamase expression by upregulating its transcription level, which increases the energy consumption of bacteria and affects its growth rate [45]. Mutations of the EF-G encoding gene *fusA* also decrease the growth rate of bacteria by reducing the protein synthesis rate [46]. Similarly, mutations that affect intracellular antibiotic levels can also increase bacterial metabolic costs. Mutations related to outer membrane porins that reduce antibiotic permeability can also block the absorption of other beneficial compounds, including nutrients [47]. In addition to consuming extra energy, the activated efflux pumps can also export beneficial compounds out of the cell, ultimately increasing the metabolic burden of bacteria [48,49]. We speculate that this is why Δ*relA*-resistant strains have not evolved mutations related to mechanisms such as antibiotic permeability and efflux pumps because these mutations will cause the already nutrient-starved strains to become even more nutrient deficient.

During later stages of resistance evolution, almost all strains evolved mutations related to antibiotic permeability and efflux pumps. The final resistant strains all had significantly lower biomass yield on glucose compared to their ancestor. Moreover, the biomass yield of Δ*relA*-resistant strains was significantly lower than that of other resistant strains. This further indicates that, in the *relA* knockout, the ability to respond to stress is weakened due to limited synthesis of (p)ppGpp, resulting in a reduced growth rate and increased energy requirement to maintain reproduction. Furthermore, we found a slight increase in the growth rate of specific resistant strains during the later stages of amoxicillin, enrofloxacin, and kanamycin evolution. At this point, we found mutations that have not been reported to be directly associated with antibiotic resistance; therefore, we hypothesize that this is the result of compensatory evolution, i.e., compensating for the metabolic burden caused by antibiotic-resistant adaptations [50,51]. Compensatory evolution allows bacteria to maintain antimicrobial-resistant properties, which is one of the reasons why resistant strains are difficult to completely eliminate [4,52,53].

## 4. Materials and Methods

### 4.1. Bacterial Strains, Growth Media, and Culture Conditions

This study employed K-12 derivatives of *E. coli*, including the Keio Knockout strains Δ*hipA*, Δ*hipB*, Δ*relA*, Δ*rpoS*, Δ*sodA*, Δ*sodB*, Δ*soxR*, and Δ*soxS*, as well as the almost identical wild-type MG1655 strain. Strains were cultured at 37 °C with shaking at 200 rpm in a phosphate-buffered minimal medium containing 100 mM NaH_2_PO_4_ and supplemented with 55 mM glucose [54].

### 4.2. MIC Determination

First, the MICs of the ancestral strains were measured. In a 96-well plate, strains with an initial OD_600_ of 0.05 were cultured in 150 μL of medium containing antibiotics at concentrations ranging from 0.5 to 4096 μg/mL in multiples of 2. The incubations took place in a spectrophotometer plate reader (Thermo Fisher Scientific; Waltham, MA USA) maintained at 37 °C with shaking. After 24 h, the MIC was defined as the lowest antibiotic concentration, at which the OD_600_ measurement value was below 0.2.

Subsequently, the ancestral strains of each bacterium were cultured in 5 mL tubes with the starting OD_600_ of 0.1 containing one of the four antibiotics (amoxicillin, enrofloxacin, kanamycin, and tetracycline) at one-quarter MIC. Through resistance evolution experiments, each strain gradually acquired resistance [14,15]. Specifically, after overnight incubation, if the OD_600_ of the culture containing antibiotics exceeded 75% of the OD_600_ of the culture without antibiotics, the bacterial strains from the antibiotic-containing culture were introduced into two fresh medium test tubes, making a starting OD_600_ of 0.1. One tube contained double the original antibiotic concentration, while the other retained the initial antibiotic concentration. After another day of incubation, if the OD_600_ of the culture with the higher antibiotic concentration surpassed 75% of the OD_600_ of the culture with the lower concentration, the bacterial culture with the higher antibiotic concentration was chosen. Otherwise, the culture with the lower antibiotic concentration was selected. This process was iteratively continued until the antibiotic concentration could no longer be increased. Throughout the resistance evolution, strains that developed increased antibiotic concentrations had their MICs measured using the 96-well plate method.

### 4.3. Growth Rate Measurements

During the evolution of resistance, strains measuring MIC were also tested for their growth rates in antibiotic-free conditions. The ancestor and resistant strains were cultured in 150 μL of medium with an initial OD_600_ of 0.05 in 96-well plates. Bacteria strains were grown overnight in a spectrophotometer plate reader with shaking, and absorbance values were recorded every ten minutes. Growthrates-in-R (https://github.com/Pimutje/Growthrates-in-R/releases/tag/Growthrates (accessed on 7 October 2023)) was used to plot growth curves and calculate growth rates. Each sample had at least three replicates. Data are presented as means ± SD; statistical significance was determined using a one-way ANOVA, * *p* < 0.05 ** *p* < 0.01, *** *p* < 0.001.

### 4.4. Biomass Yield on Glucose Measurements

Glucose consumption was quantified using High-Performance Liquid Chromatography (HPLC) [55]. The ancestor and resistant strains were cultured in 5 mL of medium with an initial OD_600_ of 0.05 in test tubes, allowing growth until the OD_600_ reached 1.0. Subsequently, the supernatant was obtained by centrifuging at 12,000 rpm for 10 min and filtered through 0.22 μm filters to prepare the samples. To make the calibration curve, glucose standards were prepared, ranging from 0 to 50 mM. These standards, along with the samples, were analyzed using an HPLC instrument (LC-20AT, Prominence, Shimadzu; Kyoto, Japan), which was equipped with a 300 × 7.8 mm Ion exclusion Rezex ROA Organic Acid H+ (8%) column (Phenomenex), a UV detector (SPD-20 A, 210 nm), and a refractive index detector (RID-20 A, 40 °C). The mobile phase consisted of 5 mM aqueous H_2_SO_4_, with a flow rate of 0.5 mL/min, and the analysis was conducted at 55 °C. The correlation of OD_600_ to cell dry weight (DW) was CDW = 0.56 × OD_600_ g/L [56]. The biomass yield on glucose was calculated using 0.56 g L^−1^/180.156 g mol^−1^ × Glucose consumption mol L^−1^ (gDW/gGlu). Each sample had at least three replicates. Data are presented as means ± SD; statistical significance was determined using a one-way ANOVA, * *p* < 0.05 ** *p* < 0.01, *** *p* < 0.001.

### 4.5. Whole Genome Sequencing

Genomic DNA extracted from populations at the intermediate stages of amoxicillin (MIC = 128 µg/mL) and kanamycin (MIC = 256 µg/mL) resistance evolution, and from populations of the final resistant strains and their respective ancestral strains, was processed using the DNeasy Blood and Tissue Kit (Qiagen Benelux B.V.; Venlo, The Netherlands). Subsequently, whole-genome sequencing was conducted, employing the NextSeq 550 next-generation sequencing system (Illumina The Netherlands B.V.; Eindhoven, The Netherlands). Sequencing analysis adhered to established protocols [57], including read alignment to reference genomes via Bowtie2. Variant calling was executed, employing Freebayes and Lofreq, with Snpeff facilitating variant annotation. Shared mutations between the resistant strains and their ancestral counterparts were excluded from the analysis. Specific single nucleotide polymorphisms (SNPs) and small insertions/deletions (indels) were documented for further investigation.

### 4.6. Quantification and Statistical Analysis

Statistical analysis employed IBM SPSS software (version 29). Specific statistical methods for each experiment are available in the figure legends and respective methods section. 

### 4.7. Data Availability

The binary alignment/map (BAM) files of the whole gene sequencing raw data have been archived in the NCBI database and are available for access through the BioProject PRJNA1047074 (MG1655) and PRJNA1047531 (Mutants).

## 5. Conclusions

During the evolution of resistance, the increase in MIC correlated with a decrease in growth rate. Furthermore, in the middle stages of amoxicillin and kanamycin resistance evolution, resistant strains with reduced growth rates showed decreased biomass yield on glucose. In addition to the general decrease, we identified two distinct growth-limiting patterns that increase this effect through genetic mutation analysis during resistance development. First, the elevated mutation rate induced by oxidative stress in the ROS scavenging-related gene knockout strains leads to the early emergence of non-specific resistance mutations, and these mutations are related to antibiotic permeability and pumping, which increase the physiological burden. Second, upon the knockout of the (p)ppGpp synthase gene *relA*, stringent response regulation became constrained, resulting in lower growth rates as a result of amino acid starvation. Genetic mutations not directly linked to antibiotic resistance are likely involved in compensatory evolution, ultimately contributing to the recovery of bacterial growth rates. This study shows how the stringent response and oxidative stress responses increase metabolic burdens, thereby reducing growth rates during the development of resistance, indicating fitness costs associated with resistance development.

## Figures and Tables

**Figure 1 ijms-25-02582-f001:**
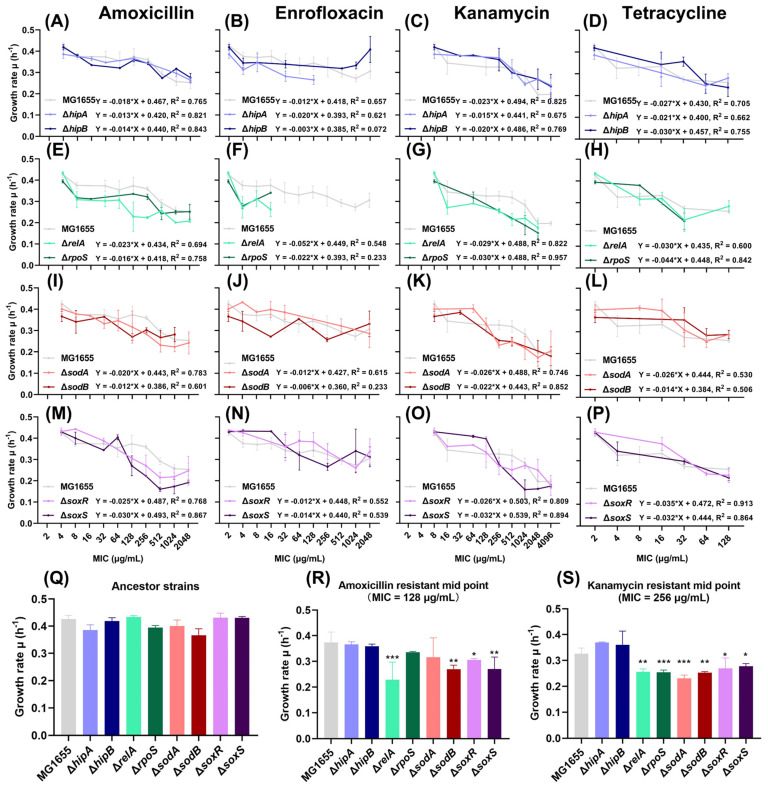
Growth rates of ROS- or (p)ppGpp-related single-gene knockout strains during resistance evolution: (**A**–**P**) The growth rates of ROS- or (p)ppGpp-related amoxicillin- (**A**,**E**,**I**,**M**), enrofloxacin- (**B**,**F**,**J**,**N**), kanamycin- (**C**,**G**,**K**,**O**), and tetracycline- (**D**,**H**,**L**,**P**) resistant evolution at each minimum inhibitory concentration (MIC). The x-axis represents the MIC, while the y-axis represents the growth rate. The linear regression equation between the log of MIC and the growth rate and R squared (R^2^) of each mutant are shown in each figure. Data are presented as means ± SD, N ≥ 3. (**Q**–**S**) The growth rate comparisons between the ancestor single-gene knockout strains (**Q**), the middle resistance evolution point (MIC = 128 µg/mL) of amoxicillin-resistant strains (**R**), and the middle resistance evolution point (MIC = 256 µg/mL) of kanamycin-resistant strains (**S**). Data are presented as means ± SD, statistical significance was determined using a one-way ANOVA, N ≥ 3, * *p* < 0.05, ** *p* < 0.01, *** *p* < 0.001.

**Figure 2 ijms-25-02582-f002:**
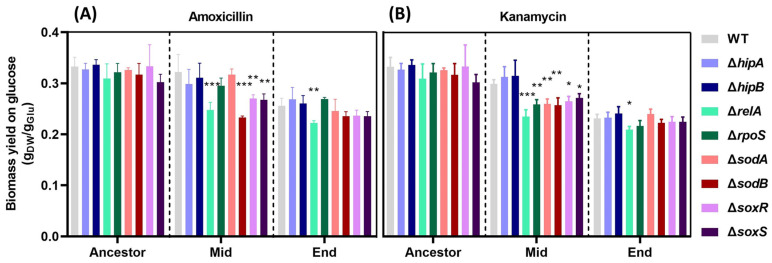
The biomass yield on the glucose of the ancestor strains and amoxicillin- or kanamycin-resistant strains. The biomass yield on the glucose of amoxicillin-resistant (**A**) and kanamycin-resistant (**B**) strains at the start, middle, and final resistance evolution points in an antibiotic-free medium. Data are presented as means ± SD; statistical significance was determined using a one-way ANOVA, N = 3, * *p* < 0.05 ** *p* < 0.01, *** *p* < 0.001.

**Figure 3 ijms-25-02582-f003:**
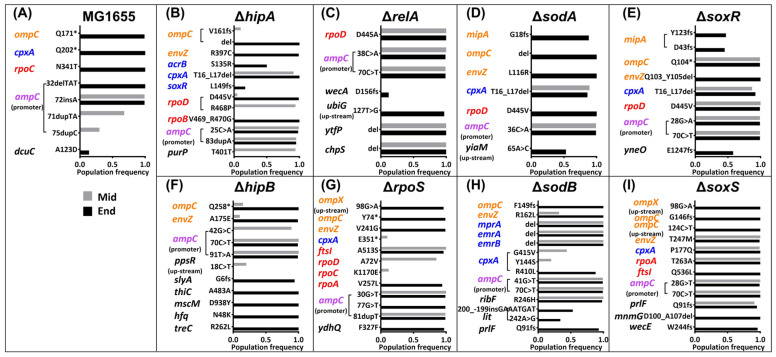
(**A**–**I**) Mutations in the amoxicillin-resistant strains during the resistance evolution. The mutated genes and their mutations at the middle (gray colors) resistance evolution point and the end (black colors) resistance evolution point during the resistance development. * indicates no amino acid change. The x-axis represents the population frequency, while the y-axis indicates the mutations. The different colors of the mutated genes represent different functions according to the Comprehensive Antibiotic Resistance Database and UniProt. Orange color means genes associated with reduced antibiotic permeability; blue color signifies genes associated with antibiotic efflux pumps; red color indicates genes associated with antibiotic target alteration; purple color means genes associated with antibiotic inactivation, and black color means genes not directly associated with antibiotic resistance.

**Figure 4 ijms-25-02582-f004:**
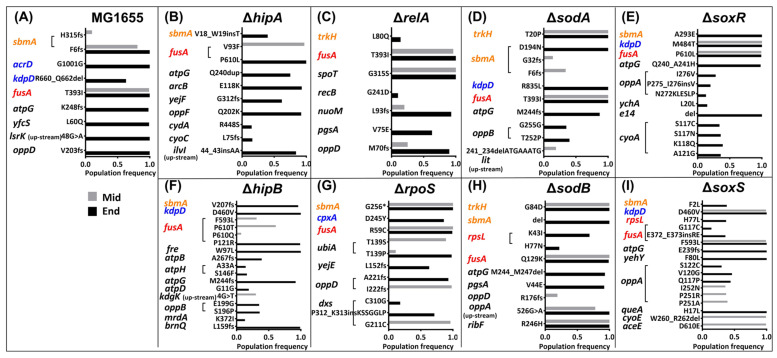
(**A**–**I**) Mutations in the kanamycin-resistant strains during the resistance evolution. The mutated genes and their mutations at the middle (gray colors) resistance evolution point and the end (black colors) resistance evolution point during the resistance development. The x-axis represents the population frequency, while the y-axis indicates the mutations. * indicates no amino acid change. The different colors of the mutated genes represent different functions according to the Comprehensive Antibiotic Resistance Database and UniProt. Orange color indicates genes associated with reduced antibiotic permeability; blue color means genes associated with antibiotic efflux pumps; red color denotes genes associated with antibiotic target alteration, and black color represents genes not directly associated with antibiotic resistance.

**Figure 5 ijms-25-02582-f005:**
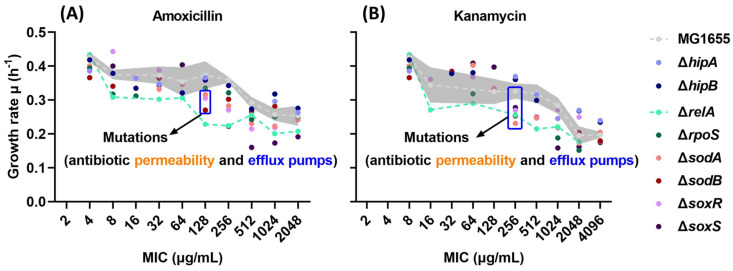
Schematic diagram of two processes causing reduced growth rates as resistance increases. As the MICs for amoxicillin (**A**) and kanamycin (**B**) increase, the growth rates of all strains gradually decrease. This relates to two different processes. ROS-related knockout strains develop mutations related to antibiotic permeability and efflux pumps earlier (in the mid-stage) than other strains. Because these mutations increase energy consumption, the fitness cost increases and the growth rate decreases. The other process is that, in the early stages of resistance evolution, Δ*relA* shows a decrease in growth rate, which may be related to it simulating starvation-induced growth arrest. Different colored dots represent the median of the growth rate of each strain at different MICs. Gray shading indicates the range of variation in WT growth rates. The green line connects the median of the growth rates of Δ*relA*.

**Table 1 ijms-25-02582-t001:** Number of times that genes mutated more than once appear in the 9 amoxicillin-resistant strains of Figure 3.

Times	2	3	4	5	6	7	8	9
Mid		* ompC * * envZ *	* rpoD *		* cpxA *			* ampC *
End	*ompX* *mipA* *rpoA* *ftsI* *prlF*		* rpoD *		* cpxA *	* envZ *	* ompC *	* ampC *

**Table 2 ijms-25-02582-t002:** The number of times that genes mutated more than once appears in the 9 kanamycin-resistant strains shown in Figure 4.

Times	2	3	4	5	6	7	8	9
Mid	* trkH * * kdpD * *oppA*	* sbmA * *oppD*						* fusA *
End	* rpsL * *oppB* *pgsA*	* trkH * *oppA* *oppD*		* kdpD *		*atpG*	* sbmA *	* fusA *

## Data Availability

Data is contained within the article.

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
