# Peer review of "The Effect of the Stringent Response and Oxidative Stress Response on Fitness Costs of De Novo Acquisition of Antibiotic Resistance"

_ijms, 2024, doi:10.3390/ijms25052582_

Round 1
Reviewer 1 Report (Previous Reviewer 2)
Comments and Suggestions for Authors
This manuscript by Qi et al. merits consideration for publication in IJMS. Although this is a resubmission, I find that this manuscript has something to offer. It is clear that exposing E. coli to antibiotics will eventually lead to resistance and selection of resistance, however the authors use a cutoff (presence of mutations >2 in the whole genome, Figs 3 and 4 Tables 1 and 2), with reproducible patterns with different antibiotics. Although not a revolutionary paper, the authors do answer their research questions.
However, first, I would like the authors comments on their static cultures for measurement of growth, would they have seen any differences using shaking cultures? One can argue that there is more oxygen perfused in shaking cultures, would they have expected a more drastic effect of ROS? Perhaps this can be a follow up experiment or if they have time they can include it in this manuscript.
Second, I am surprised that the relA mutant ancestor strain did not have any growth difference, are these cultures always in log phase? One can predict differences in growth at the plateau phase, which inherently is tied to the log phase of growth. The authors should comment on this.
Author Response
Reviewer 1
This manuscript by Qi et al. merits consideration for publication in IJMS. Although this is a resubmission, I find that this manuscript has something to offer. It is clear that exposing E. coli to antibiotics will eventually lead to resistance and selection of resistance, however the authors use a cutoff (presence of mutations >2 in the whole genome, Figs 3 and 4 Tables 1 and 2), with reproducible patterns with different antibiotics. Although not a revolutionary paper, the authors do answer their research questions.
However, first, I would like the authors comments on their static cultures for measurement of growth, would they have seen any differences using shaking cultures? One can argue that there is more oxygen perfused in shaking cultures, would they have expected a more drastic effect of ROS? Perhaps this can be a follow up experiment or if they have time they can include it in this manuscript.
This comment was slightly confusing for us. All incubations were performed with shaking. This is now more clearly mention in the M&M section. We do agree that not shaking would cause an oxygen gradient to build up, with unknown consequences. However, we don’t have any data on this as all incubation were performed with shaking.
Second, I am surprised that the relA mutant ancestor strain did not have any growth difference, are these cultures always in log phase? One can predict differences in growth at the plateau phase, which inherently is tied to the log phase of growth. The authors should comment on this.
Indeed, ΔrelA had a similar growth rate as the WT during the log phase. The growth of ancestor relA knockout strain is regular when not exposed to antibiotics, but decreases when exposed to non-lethal concentrations. When exposed to antibiotics, bacteria need to synthesize additional proteins which causes deacylated tRNA to appear which in turn hinders transcription elongation. The RelA response to deacylated tRNA then produces (p)ppGpp. This compound regulates the transcription and amino acid biosynthesis, turning deacylated tRNA into amino-acylated tRNA to maintain growth. This is now explained in lines 247-257.
Reviewer 2 Report (New Reviewer)
Comments and Suggestions for Authors
The authors describe a study on acquisition of antibiotics and indicate to study the role of oxidative stress response and stringent response. However, the study has serious flaws, which are not easily solved. These flaws come down to the general subjects addressed below.
The research design is not appropriate. There is an indication that the acquisition of antibiotics experiments have been performed in multiple replicates. But a clear indication is not given. it indicates to be 3 or more. For which strain was what number of replicates performed? Moreover, for the sequencing that was performed it is not indicated if this was performed for replicates. and how often a gene per strain was mutated.
Also the authors indicate that a culture after acquisition of the resistance is a strain. But as it is a culture, and many different strains can have started to exist, it should be regarded as a community. Sequencing analysis of a community should be done differently than analysis of a strain as it is not whole genome sequencing. In figure 3, there are indications of population frequency, but this not addressed in the text.
The methods are not adequately described. The culturing experiments really need to be described in more detail. The amount that is transferred needs to be added.
Also the Wildtype strains differs a bit from the mutant strains, and as this study focusses on SNPs these differences need to be discussed and described.
Results are not clearly presented. as indicated it is not clear for instance how often a mutation is found per mutated strain. it is not clear if there are mutations that are specific for oxidative stress, or stringent response or are general. it seems that there are no specific mutations, however the authors present that there are differences and specifics per group. If these specific differences are there (which is thus doubtful from the now presented results) they need to be presented more clearly.
MICs of the cultures were measured, but are not reported. It is needed to see if the mutations in a low antibiotic concentration already results in a resistance to higher concentrations.
The conclusions are not supported by the results. As indicated above the results are not presented clearly. And from the results, as they are now presented, it is not clear that there is a specific mutation or specific mutations that can be related to the mutant strains studied. The results and conclusions are supported by too much speculative discussion. These speculative discussions are used for the conclusions, but this is way overdone.
As these major flaws need to be addressed, more detailed comments are for now too early.
Author Response
Please see the attached file

Reviewer 3 Report (New Reviewer)
Comments and Suggestions for Authors
In this paper the authors study the Effect of the Stringent Response and Oxidative Stress Response on Fitness Costs of de novo Acquisition of Antibiotic Resistance. The paper is very well written and the results are well presented. The introduction is quite long and can be shortened.
Author Response
Reviewer 3
In this paper the authors study the Effect of the Stringent Response and Oxidative Stress Response on Fitness Costs of de novo Acquisition of Antibiotic Resistance. The paper is very well written and the results are well presented. The introduction is quite long and can be shortened.
The introduction is indeed long, but since the subject is unusual and the study itself rather intricate we needed to provide the reader with more background than usual.
Round 2
Reviewer 1 Report (Previous Reviewer 2)
Comments and Suggestions for Authors
Thank you for addressing my concerns. The manuscript is intellectually stimulating.
Author Response
Thank you for this comment
Reviewer 2 Report (New Reviewer)
Comments and Suggestions for Authors
I will start with an apology, as I made the mistake of not typing "antibiotic resistance" in the indication of what the paper is about. that does of course not help in convincing you that the points I address need to be really be improved.
And actually they all still stand, except for a small improvement in the methods and the addition that we are looking at population frequency. Actually the rebuttal convinced me of some points that I indicated to be wrong.
As I understand from the rebuttal the "resistance acquisition" experiment was performed only once for each starting-strain. This is a major flaw of the study. There is no indication that this experiment is representative for repetitions. So conclusions can certainly not been drawn.
The authors in the rebuttal state: "We show that mutagenic effects of ROS cause mutations in genes related to antibiotic permeability and efflux pumps."
This is not correct. You show that some genes are mutated, and other studies have studied their function. But the role in antibiotics permeability and efflux is often speculative. The authors state them self: These mutations may be related to reduced antibiotic permeability [28][29].
However, in the conclusion all the may and possible have been changed into statements. This should stay speculative and stated as not proven.
Also from the rebuttal: "This observation is presented using the pattern diagram in Figure 5 and explained in the discussion."
The figure 5 points shows (from the n=1 experiment) that ROS-related knockouts have a lower growth rate at middle MICs. But it does not show that there are specific genes mutated in the ROS-related knockouts.
Only for kanamycin I can see kdpD maybe is specific to ROS- related KO's. but not very convincing, as it is still seen later in WT and hipB, and not in the other two ROS-related KO's. And trkH seems specific for the sod mutants.
But this is all with keeping in mind that the resistance acquisition experiment was performed n=1 for all these strains.
the authors still indicate the communities as strains, and call the sequencing of the communities whole genome sequencing. This is not correct.
The description on the method of the sequencing analysis needs to be improved. it is a major part of this manuscript and needs to be written out, in stead of referring to others. (and should not indicate whole genome sequencing)
Comments on the Quality of English Language
some minor issues. please read again carefully (nut i.s.o. not and some other small things).
Round 3
Reviewer 2 Report (New Reviewer)
Comments and Suggestions for Authors
I am surprised about the rebuttal and even a bit shocked. the authors have not even taken the effort to improve the typo indicated ("nut" in line 341).
Most surprising is the way they look at how to interpret results. For me that is not the way we decided to do things in science. Results that are indications or suggestions for conclusions should be presented as such, and not as hard conclusions. And I believe that is the way we in science decided to work with each other. Even if the results are indications these can still be of value, but it must be clearly presented as indications. These can then be used to further hypothesize.
Confusing from the rebuttal is the fact that in my review I indicated it seems that the resistance acquisition has not been performed in replicates. The authors do not respond to this remark. But in the answer to another remark they now indicate the acquisition experiment has been performed in triplicate. This is really strange as this was already an indication in the first review report. If there are replicates this should be clearly shown and indicated in the paper.
The authors indicate that previous papers have shown that replicates are almost always the same and therefore also here sequencing does not have to be performed in replicates. Also this is not how it was decided to do things in science. Replicates are always necessary for hard conclusions. And in this case it has never been shown that the mutant-strains act the same in replicate experiments, so also in this case replicates should be performed.
Still also remains the fact that the results that lead to the conclusion are not presented in a way that you can make this out from the figures. this has to be pointed out much better (and I believe that if this is done, it is actually clear that the results cannot lead to the strong conclusions given now).
Author Response
Please see the attached file

This manuscript is a resubmission of an earlier submission. The following is a list of the peer review reports and author responses from that submission.
Round 1
Reviewer 1 Report
Comments and Suggestions for Authors
The authors present a study on the impact of antimicrobial resistance in E. coli mutants related to the stringent response/oxidative stress. While increasing resistance to antimicrobials is a concern in the current healthcare context, it is unclear what novelty this manuscript brings to the field. The fact that most strains showed decreases in growth rate/biomass is not surprising as this is well established in the field and these two factors relate to each other (decreased growth rate would naturally lead to lower biomass).
The authors also reference an older study suggesting that most antibiotics work through a common ROS pathway. This study has been shown to have several methodological flaws, with rebuttal papers presenting opposing results published in Science (10.1126/science.1232688;10.1126/science.1232751). The role of ROS in antibiotics is thus controversial.
There is also no validation of the results; none of the mutations identified were tested for their impact on antibiotic resistance/growth; without this is is unclear how many of these mutations are simply random fixations occurring due to repeated rounds of selection. Indeed, the authors simply assume that a mutation in a gene previously known to be involve in antibiotic resistance (e.g. ampC/fusA) will impact the gene in the same manner as other studies without any demonstration that this is actually occuring. In addition, the sequencing data for enrofloxacin/tetracycline are not shown/presented.
The raw sequencing data for these experiments should be also submitted to a publicly available repository.
Author Response
IJMS-2735638 Reviewer 1
The authors present a study on the impact of antimicrobial resistance in E. coli mutants related to the stringent response/oxidative stress. While increasing resistance to antimicrobials is a concern in the current healthcare context, it is unclear what novelty this manuscript brings to the field.
The novelty of this study that is we were able to elucidate the roles of the stringent response and oxidative stress responses. Understanding their impact on resistance acquisition and fitness costs targets vulnerabilities in resistance development, specifically the stringent response and oxidative stress This information can be used to develop novel strategies to reduce resistance development.
The fact that most strains showed decreases in growth rate/biomass is not surprising as this is well established in the field and these two factors relate to each other (decreased growth rate would naturally lead to lower biomass).
Again, the new insight is not the general principle, but the differences between antibiotics and between the wild-type and the various mutants. That analysis provides new understanding of the metabolic adaptations in the cell, as e.g. demonstrated by the lower growth rates of ROS- or (p)ppGpp-related single-gene knockout strains than WT at mid-evolution compared to the final stages.
The authors also reference an older study suggesting that most antibiotics work through a common ROS pathway. This study has been shown to have several methodological flaws, with rebuttal papers presenting opposing results published in Science (10.1126/science.1232688;10.1126/science.1232751). The role of ROS in antibiotics is thus controversial.
This dispute has indeed been long running, but all our recent studies are best explained in the framework of the radical based theory. This year we published already two articles that contribute to this controversy : https://doi.org/10.1186/s12866-023-03031-4 and https://doi.org/10.1016/j.isci.2023.108373 . We did not want to double publish by repeating those arguments. That does not mean that we claim to have proven the theory, only that our observations are best explained by it.
There is also no validation of the results; none of the mutations identified were tested for their impact on antibiotic resistance/growth; without this is is unclear how many of these mutations are simply random fixations occurring due to repeated rounds of selection.
Tables 1 and 2 answer this question in an indirect manner. We feel it is safe to assume that a mutation that appears in 9 different strains fulfils an important function, while those that appear fewer times have correspondingly less important roles. In our studies on this subject we have found some 150 “usual suspects” and thousands of mutation that seem to occur randomly. Even with 150 mutations, to create these mutants and perform all the necessary experiments would require an enormous amount of work, with as only outcome the confirmation of an earlier conclusion.
Indeed, the authors simply assume that a mutation in a gene previously known to be involve in antibiotic resistance (e.g. ampC/fusA) will impact the gene in the same manner as other studies without any demonstration that this is actually occuring. In addition, the sequencing data for enrofloxacin/tetracycline are not shown/presented.
The analysis of the data for amoxicillin and kanamycin already took considerable space in the text. Tetracycline sequencing would not have given much additional insight as it is a bacteriostatic antibiotic and in the case of the fluoroquinolones this analysis of the mutation in gyrA, gyrB and parC, etc has been performed many times before by our group and others.
The raw sequencing data for these experiments should be also submitted to a publicly available repository.
Accepted. We submitted it to NCBI and added the information in the manuscript.
Reviewer 2 Report
Comments and Suggestions for Authors
This study by Qi et al. merits consideration for publication in IJMS. It involves a highly important subject, antibiotic resistance, in the Gram-negative bacterium E. coli, as well as mutants of E. coli in the stringent and ROS response (e.g., deltarelA, and deltasoxR-S, to highlight a few). The experiments conducted are sound, however, one note the authors should add is whether the E. coli and mutants in both responses ever reached the plateau phase under their experimental conditions, especially the MIC versus growth rate correlations (Fig.1)? A growth curve could be useful here, as the stringent response is notably turned on in the plateau phase, which could serve as a confounder. If this phase was never achieved, they should state it. Why did they choose to grow their bacteria in the phosphate buffer minimal media and not N-minimal media (https://www.sciencedirect.com/science/article/pii/S0092867400000921)? Also, please state whether magnesium was added to the media, as this has great influence on membrane permeability, especially Gram-negative bacteria. Given the authors use a minimal medium in their experiments, it could be good to comment on the baseline activation of the stringent response in this medium.
A few minor points include: line 47, it is suggested not to use "reproductive" here, use "replicative".
Also, in the introduction, the authors may think to highlight Gram-negative antimicrobial resistance, which is a growing problem, instead of AMR in general.
Author Response
IJMS-2735638 Reviewer 2
This study by Qi et al. merits consideration for publication in IJMS. It involves a highly important subject, antibiotic resistance, in the Gram-negative bacterium E. coli, as well as mutants of E. coli in the stringent and ROS response (e.g., deltarelA, and deltasoxR-S, to highlight a few).
The experiments conducted are sound, however, one note the authors should add is whether the E. coli and mutants in both responses ever reached the plateau phase under their experimental conditions, especially the MIC versus growth rate correlations (Fig.1)?
Accepted and corrected on line 179-180.
A growth curve could be useful here, as the stringent response is notably turned on in the plateau phase, which could serve as a confounder. If this phase was never achieved, they should state it.
Accepted, see above this statement is on lines 179-180.
Why did they choose to grow their bacteria in the phosphate buffer minimal media and not N-minimal media (https://www.sciencedirect.com/science/article/pii/S0092867400000921)?
The medium for the reference was for Salmonella spp. The medium we used is a very standard E. coli minimal medium that we have used in all our studies over the past 15-plus years.
Also, please state whether magnesium was added to the media, as this has great influence on membrane permeability, especially Gram-negative bacteria.
The medium contains 1.25 mM MgCl2•6H2O.
Given the authors use a minimal medium in their experiments, it could be good to comment on the baseline activation of the stringent response in this medium.
This point is addressed in the discussion on lines 332 – 344.
A few minor points include: line 47, it is suggested not to use "reproductive" here, use "replicative".
Accepted and corrected accordingly.
Also, in the introduction, the authors may think to highlight Gram-negative antimicrobial resistance, which is a growing problem, instead of AMR in general.
Accepted; corrected in line 35.
Round 2
Reviewer 1 Report
Comments and Suggestions for Authors
The authors present a study on the impact of antimicrobial resistance in E. coli mutants related to the stringent response/oxidative stress. While increasing resistance to antimicrobials is a concern in the current healthcare context, it is unclear what novelty this manuscript brings to the field.
The novelty of this study that is we were able to elucidate the roles of the stringent response and oxidative stress responses. Understanding their impact on resistance acquisition and fitness costs targets vulnerabilities in resistance development, specifically the stringent response and oxidative stress This information can be used to develop novel strategies to reduce resistance development.
I fail to see how the information gleaned in this study would lead to the development of strategies to reduce resistance development. Most pathogens require functional oxidative stress/stringent responses in order to cause disease (regardless of antibiotic presence), so the impact of their deletion on antibiotic resistance would seem to be moot as these strains would not be able to colonize/cause disease in the first place.
The fact that most strains showed decreases in growth rate/biomass is not surprising as this is well established in the field and these two factors relate to each other (decreased growth rate would naturally lead to lower biomass).
Again, the new insight is not the general principle, but the differences between antibiotics and between the wild-type and the various mutants. That analysis provides new understanding of the metabolic adaptations in the cell, as e.g. demonstrated by the lower growth rates of ROS- or (p)ppGpp-related single-gene knockout strains than WT at mid-evolution compared to the final stages.
There is no explanation or mechanism proposed to explain why these strains would show decreased growth rates at mid-range resistance levels. Nor it is clear why the fact that mid-levels of antibiotic resistance and growth rates in these mutants has any clinical/practical relevance.
The authors also reference an older study suggesting that most antibiotics work through a common ROS pathway. This study has been shown to have several methodological flaws, with rebuttal papers presenting opposing results published in Science (10.1126/science.1232688;10.1126/science.1232751). The role of ROS in antibiotics is thus controversial.
This dispute has indeed been long running, but all our recent studies are best explained in the framework of the radical based theory. This year we published already two articles that contribute to this controversy : https://doi.org/10.1186/s12866-023-03031-4 and https://doi.org/10.1016/j.isci.2023.108373 . We did not want to double publish by repeating those arguments. That does not mean that we claim to have proven the theory, only that our observations are best explained by it.
There is a difference between the theory that antibiotics are commonly using ROS as a secondary means to kill their targets vs the presence of ROS increasing the rates of mutations that lead to antibiotic resistance. It is well known that mutants in DNA repair enzymes (e.g. mutS) are able to develop resistance to antibiotics at a faster rate than WT strains, but this does not mean that antibiotics work though a common pathway of involving DNA repair. This distinction should be made more apparent.
There is also no validation of the results; none of the mutations identified were tested for their impact on antibiotic resistance/growth; without this is is unclear how many of these mutations are simply random fixations occurring due to repeated rounds of selection.
Tables 1 and 2 answer this question in an indirect manner. We feel it is safe to assume that a mutation that appears in 9 different strains fulfils an important function, while those that appear fewer times have correspondingly less important roles. In our studies on this subject we have found some 150 “usual suspects” and thousands of mutation that seem to occur randomly. Even with 150 mutations, to create these mutants and perform all the necessary experiments would require an enormous amount of work, with as only outcome the confirmation of an earlier conclusion.
I disagree that Table 1&2 demonstrate whether these mutations may be involved in resistance. Firstly, your statement that “a mutation that appears in 9 different strains fulfills an important function” is misleading as essentially none of the mutations are conserved across strains, only the gene in which you identified a mutation. It is a leap to assume that all of the mutations that were identified would have a direct (and similar) impact on antibiotic resistance without some kind of testing. Indeed, the genes which appear in all 9 strains (ampC/fusA) have been previously reported to be involved in antibiotic resistance so their presence is unsurprising. Even we can assume that every mutation identified in ampC/fusA improved antibiotic resistance there is no evidence that any of the other mutations identified play any role whatsoever. There is no need to exhaustively mutate and confirm each and every mutation identified, but there is currently no validation whatsoever.
You also state above that there are “thousands of mutation that seem to occur randomly”, this data does not appear to be presented or explained why you focused on the mutations listed in the table. What was the criteria for choosing a mutation to include in Tables 1&2?
Indeed, the authors simply assume that a mutation in a gene previously known to be involve in antibiotic resistance (e.g. ampC/fusA) will impact the gene in the same manner as other studies without any demonstration that this is actually occuring. In addition, the sequencing data for enrofloxacin/tetracycline are not shown/presented.
The analysis of the data for amoxicillin and kanamycin already took considerable space in the text. Tetracycline sequencing would not have given much additional insight as it is a bacteriostatic antibiotic and in the case of the fluoroquinolones this analysis of the mutation in gyrA, gyrB and parC, etc has been performed many times before by our group and others.
The rational for not doing the genomic sequencing for these antibiotics should be better explained in the main text. It is not clear why a bacteriostatic antibiotic would yield different results in your genomic sequencing from a bactericidal antibiotic (if so, why did you study the growth rates etc). One could also argue that WT strains have been sequenced in the past for penicillin/aminoglycoside antibiotics as well so it is unclear why fluoroquinolones are specifically being excluded (these previous sequencing studies should also be referenced/discussed).
The raw sequencing data for these experiments should be also submitted to a publicly available repository.
Accepted. We submitted it to NCBI and added the information in the manuscript.
Thank you for submitting the raw data.